**Data Availability Statement:** The dataset from this study is held securely in coded form within Alberta

# Comparison of hospitalization events among residents of assisted living and nursing homes during COVID-19: Do settings respond differently during public health crises?

Colleen J. Maxwell[1,2]*, Eric McArthur[3], David B. Hogan[4], Hana Dampf[5], Jeffrey Poss[6], Joseph E. Amuah[7], Susan E. Bronskill[2,8], Erik Youngson[9,10], Zoe Hsu[9,10], Matthias Hoben[5,11]

1 School of Pharmacy, University of Waterloo, Waterloo, Ontario, Canada, 2 ICES, Toronto, Ontario, Canada, 3 London Health Sciences Centre, London, Ontario, Canada, 4 Division of Geriatric Medicine, Cumming School of Medicine, University of Calgary, Calgary, Alberta, Canada, 5 Faculty of Nursing, College of Health Sciences, University of Alberta, Edmonton, Alberta, Canada, 6 School of Public Health Sciences, University of Waterloo, Waterloo, Ontario, Canada, 7 School of Epidemiology and Public Health, University of Ottawa, Ottawa, Ontario, Canada, 8 Dalla Lana School of Public Health, University of Toronto, Toronto, Ontario, Canada, 9 Provincial Research Data Services, Alberta Health Services, Alberta, Canada, 10 Data and Research Services, Alberta SPOR SUPPORT Unit, Alberta, Canada, 11 School of Health Policy and Management, Faculty of Health, York University, Toronto, Ontario, Canada

* colleen.maxwell@uwaterloo.ca

## Abstract

### Background

COVID-19 and resulting health system and policy decisions led to significant changes in healthcare use by nursing homes (NH) residents. It is unclear whether healthcare outcomes were similarly affected among older adults in assisted living (AL). This study compared hospitalization events in AL and NHs during COVID-19 pandemic waves 1 through 4, relative to historical periods.

### Methods

This was a population-based, repeated cross-sectional study using linked clinical and health administrative databases (January 2018 to December 2021) for residents of all publicly subsidized AL and NH settings in Alberta, Canada. Setting-specific monthly cohorts were derived for pandemic (starting March 1, 2020) and comparable historical (2018/2019 combined) periods. Monthly rates (per 100 person-days) of all-cause hospitalization, hospitalization with delayed discharge, and hospitalization with death were plotted and rate ratios (RR) estimated for period (pandemic wave vs historical comparison), setting (AL vs NH) and period-setting interactions, using Poisson regression with generalized estimating equations, adjusting for resident and home characteristics.

### Results

On March 1, 2020, there were 9,485 AL and 14,319 NH residents, comparable in age (mean 81 years), sex (>60% female) and dementia prevalence (58–62%). All-cause hospitalization

Health Services. While ethical restrictions and legal data sharing agreements between the investigators, the Alberta SPOR SUPPORT Unit and data providers (Alberta Health Services and Alberta Health) prohibit making the dataset publicly available, access may be granted to those who meet pre-specified criteria for access, available at: https://absporu.ca/. email: absporu@albertainnovates.ca.

**Funding:** This study was funded by a Canadian Institutes of Health Research (CIHR) Operating Grant (reference # 179467), co-held by MH & CJM. The funders had no role in the study design, data collection and analyses, decision to publish, or preparation of the manuscript.

**Competing interests:** The authors have declared that no competing interests exist.

rates declined in both settings during waves 1 (AL: adjusted RR 0.60, 95%CI 0.51–0.71; NH: 0.74, 0.64–0.85) and 4 (AL: 0.76, 0.66–0.88; NH: 0.65, 0.56–0.75) but unlike NHs, AL rates were not significantly lower during wave 2 (and increased 27% vs NH, January 2021). Hospitalization with delayed discharge increased in NHs only (during and immediately after wave 1). Both settings showed a significant increase in hospitalization with death in wave 2, this increase was larger and persisted longer for AL.

## Conclusions

Pandemic-related changes in hospitalization events differed for AL and NH residents and by wave, suggesting unique system and setting factors driving healthcare use and outcomes in these settings in response to this external stress.

## Introduction

Assisted living (AL) represents an increasingly common residential setting for older adults with health and support needs [1, 2]. Primarily aligned with a social model of care, AL homes typically provide a secure environment, personal support services and limited health care, emphasizing a home-like setting [2]. Relative to nursing homes (NHs), AL settings have lower levels of professional staffing (no or limited on-site registered nurses, physicians, nurse practitioners), fewer health and medical services and a greater emphasis on family involvement in resident care [3–6]. These features contrast with the clinical complexity of the AL population, with residents exhibiting a high prevalence of dementia, multimorbidity, frailty and complex polypharmacy [1, 2, 6, 7]. Research on the ability of AL to provide quality care to residents with changing health needs remains sparse though studies have raised concerns about insufficient staffing levels and mix [8–10], medication use and oversight [11], and high rates of care transitions, including hospital and NH admissions [2, 8, 12, 13]. As such, AL settings may be particularly sensitive to external stresses because of their limited reserve.

Hospitalization events are common and an important indicator of the quality and availability of care in AL [8, 13]. Associated risk factors include both AL resident (e.g., health instability, polypharmacy) and home characteristics (e.g., smaller settings, lower staffing levels and the absence of on-site professional staff) [8–10]. Prior to the COVID-19 pandemic, studies within the U.S. [12, 14] and Canada [8, 13, 15] showed a significantly higher rate of hospitalization (and prolonged stays with delayed discharge) for AL residents relative to the NH population. The relative absence of professional staff, medical services and integrated interprofessional care, arguably resulting in less timely and/or appropriate care (e.g., failure to address upstream risks for falls, delirium, and infections; delayed detection or treatment of acute health changes), is felt to contribute to the higher hospitalizations observed for AL [6, 16, 17]. These hospitalization events often lead to lengthy hospital stays and worsening functional and cognitive health status for residents [18] (prompting a NH admission) [2] and increased distress for family caregivers. Yet, it is important to acknowledge that many hospitalizations are necessary and that reduced access to needed hospital care (e.g., during healthcare crises) also constitutes a relevant system quality indicator for at-risk older populations [19]. Given these considerations, it is likely that the healthcare disruptions associated with the COVID-19 pandemic (e.g., staffing shortages, reduced availability of social/medical services and family to support care) were especially challenging for AL homes [3–5, 20] potentially leading to heightened risks of adverse health outcomes for residents.

Studies of older adults in community [21, 22] and nursing home [23–28] settings showed that hospitalization and emergency department rates declined dramatically during the first pandemic wave and typically remained below historical rates during the second wave [22, 27, 28]. Other than a descriptive study of hospitalizations from residential care and nursing homes in England early in the pandemic, that showed few differences between the settings [24], research has yet to rigorously examine whether the COVID-19 pandemic and different pandemic waves (that varied in case numbers, infection virulence, vaccination rates, public health restrictions, and availability of staff and services) [28] differentially affected hospitalizations among AL compared with NH residents. This is a particularly salient research question given historical patterns of hospitalizations evident for these two settings caring for vulnerable older residents.

Our objectives were to compare rates of hospitalization events (all-cause hospitalization, hospitalization with delayed discharge, hospitalization with death) among residents of AL and NH settings in Alberta, before and following the onset of the COVID-19 pandemic, and to explore variation in rates by pandemic waves 1–4 for these two settings. We hypothesized that hospitalization rates would be higher for AL than NH residents (pre- and post-pandemic onset) and there would be significant decreases in rates for all-cause and delayed hospitalizations (but increases in hospitalization with death) that would be more pronounced for NH than AL residents and vary by pandemic wave.

## Methods

### Study design, setting and data

We conducted a population-based, repeated cross-sectional study of hospitalizations among all residents of publicly subsidized assisted living (AL) and nursing home (NH) settings in Alberta Canada between January 1, 2018 and December 31, 2021. Alberta Health Services (AHS) coordinate and provide continuing care services through a single point of access, including licensed publicly subsidized AL (designated supportive living in Alberta) [29] and NH (or residential long-term care) care in the province. Starting March 20, 2020, both settings were under a Provincial Chief Medical Officer of Health order to restrict visitor access, allowing a single essential visitor only [30].

The following provincial clinical and health administrative databases were used (extracted from the AHS Enterprise Data Warehouse and provided by AHS as individual level de-identified data and linked using unique resident identifiers): Alberta Continuing Care Information System (ACCIS), capturing data on residents and homes, including the Resident Assessment Instrument–Home Care (RAI-HC) and Minimum Data Set 2.0 (RAI-MDS 2.0) used in AL and NH, respectively; Discharge Abstract Database (DAD) for inpatient hospitalizations; Provincial Laboratory for COVID-19 testing data (month/year only); Immunization & Adverse Reactions to Immunization for aggregate data on vaccine administration; and Vital Statistics database for death dates (**S1 Table**). These de-identified data were accessed for this research on the 29[th] of April, 2022 (with continued access for analyses until December 31, 2023). The RAI-HC and RAI-MDS 2.0 are validated assessments [31, 32] routinely completed on admission, after a significant health change and annually, and for NH only—on a quarterly basis. These assessments capture residents' physical, cognitive and psychosocial status, chronic conditions, and service use.

### Study population

All AL and NH residents with one or more RAI assessments completed during the study period were identified and all assessments for this period (plus those completed during a

1-year look back period) were captured. We identified residents alive and living in AL or NH on the first day of each month (monthly index date). Included were 26 historical (pre-pandemic) monthly periods (January 1, 2018 to February 29, 2020) and 22 pandemic monthly periods (March 1, 2020 to December 31, 2021). On each monthly index date, we excluded residents with a hospital stay overlapping this index date. Relevant admission, discharge, and death dates to define monthly cohorts were obtained from the ACCIS, DAD, and Vital Statistics databases.

## Measures

**Exposures.** Our main exposures were period (COVID-19 pandemic month relative to comparable historical [2018–2019 combined] month) and setting (AL vs NH). Specific focus was on months that represented peaks of the first 4 pandemic waves in Alberta (April 2020, December 2020, April 2021, September 2021) [33], a month with low count of COVID-19 cases that was prior to the start of vaccinations in AL and NH (August 2020), and a month with an observed peak in hospitalizations for AL residents (January 2021).

**Outcomes.** For each monthly period of interest, we utilized the DAD to capture all-cause inpatient hospitalizations (primary outcome), hospitalizations with a delayed discharge (termed alternate level of care [ALC] in Canada), and hospitalizations with a discharge disposition of death (secondary outcomes). An ALC designation occurs when a patient occupying a hospital bed no longer requires acute-care resources or services and is awaiting ongoing supportive care in the community, AL or NH setting [34]. Monthly outcome rates were derived with person-days as the denominator (censoring for residents' date of transfer/discharge from the setting, death or monthly end date).

For each setting, we used the DAD to examine the top 10 ICD-10-CA [35] diagnoses/chapters for the most responsible diagnosis recorded for hospitalizations during April 2020 and January 2021 (relative to comparable historical [2018–2019] months). We identified setting-specific monthly COVID-19 related hospitalizations during the pandemic period using ICD-10-CA codes (any diagnosis field) of U07.1, U07.2, or U07.3 [36].

**Resident & home characteristics.** The most recent RAI assessment before each monthly index date was used to capture residents' age, sex, scores on validated health outcomes derived from RAI items (the Activities of Daily Living Self-Performance Hierarchy Scale [ADL-H] [37], Cognitive Performance Scale [CPS] [38], Depression Rating Scale [DRS] [39], Changes in Health, End-Stage Disease, Signs and Symptoms Scale [CHESS; assessing health instability] [40]), frailty status, behaviors, and chronic health conditions. Frailty was defined using a validated frailty index (FI), derived as the proportion of accumulated to potential health deficits (72 items on the RAI), and categorized as robust (FI<0.2), pre-frail (FI 0.2–0.3) and frail (FI>0.3) [41].

For each AL and NH, four home characteristics (health zone, urban/rural location, bed size, and for-profit vs non-profit ownership) were obtained from the home's postal code, ACCIS database, and AHS continuing care registries. In Alberta, AHS is organized into five health zones to facilitate healthcare decisions and delivery.

## Analyses

Descriptive analyses compared resident characteristics between AL and NH annually (on March 1st) across study years. Standardized differences [42] of greater than 0.10 were used to illustrate meaningful differences between AL and NH populations at each time point and within each care setting over time (comparing study end date to March 1, 2019). The monthly hospitalization event rates (per 100 person-days) were plotted for historical (2018–2019 combined) and 2020–2021 pandemic periods, by setting.

For each month of interest, separate Poisson regression models with an offset term for log-transformed follow-up time and generalized estimating equations (to account for repeated measures within individuals across months) were used to estimate rate ratios (RR) (and 95% confidence intervals [CI]) for period (pandemic vs historical), setting (AL vs NH) and period-setting interactions. Models were adjusted for age, sex, CPS score, ADL-H score, CHESS, number of chronic conditions, and home health zone and ownership status (derived at the resident level to account for regional health systems and owner/operator policies).

In sensitivity analyses we (i) restricted all-cause hospitalizations to urgent (unplanned) hospitalizations only, and (ii) estimated rate ratios for all-cause hospitalizations using a generalized linear mixed model with a Poisson distribution and random effects for individuals nested within homes to account for clustering by individual and home.

We employed p <0.05 as the level of statistical significance (two-tailed) and all analyses were conducted with SAS version 9.4 (SAS Institute Inc.).

This study received ethics clearance from the University of Alberta Health Research Ethics Board (Pro00116520), University of Calgary Conjoint Health Research Ethics Board (pSite-22-0001), York University Office of Research Ethics-Human Participants Review Sub-Committee (e2022-239) and operation approval from AHS. This study was granted a waiver of informed consent as the study team received only de-identified data and had no contact with the study population. The de-identified data were accessed for this research on the 29th of April, 2022.

The study is reported per RECORD guidelines (**S1 Text**).

## Results

During the study period, there were 439,218 AL and 673,486 NH monthly records included after exclusions for deaths or hospitalizations at the time of monthly index dates (1–2% of records), representing 20,997 unique AL and 35,487 NH residents.

On March 1, 2020, there were 9,485 AL and 14,319 NH residents, representing 251 AL and 181 NH sites. The two settings were comparable in terms of resident age (mean 81 years), sex (63–67% female), number of chronic conditions (34% with 5+ conditions) and the presence of select conditions including dementia (58–62%) (**Table 1**). Relative to NH residents, those in AL showed significantly lower levels of ADL and cognitive impairment, depressive symptoms, health instability, frailty, and responsive behaviours. There were few differences in AL resident characteristics over time except for a significant decrease in the proportion of residents with meaningful depressive symptoms. Several NH resident characteristics also remained stable over time, however there were declines during the study period in the prevalence of frailty, multiple chronic conditions (5+) and specific conditions (dementia, arthritis) among residents in this setting.

AL residents were more likely than NH residents to reside in the South zone, smaller homes and those owned/operated by voluntary non-profit and private for-profit organizations (rather than by AHS or contracted non-profit operators) (**Table 2**). The urban/rural distribution of residents was comparable for both settings. COVID-19 vaccination of both AL and NH residents started January 1, 2021 and by May 28, 2021, 85% had received their second dose. By study end about 70–80% had received a third dose.

There was a statistically significant decrease in the rate of all-cause hospitalization during the peak of pandemic wave 1 (April 2020) for residents of both AL (adjRR 0.60, 95% CI 0.51–0.71) and NHs (adjRR 0.74, 95% CI 0.64–0.85) relative to historical comparisons (**Table 3**, **Fig 1 and S1 Fig**). Over subsequent months up to and including pandemic wave 3, the hospitalization rate among NH residents increased slightly but remained significantly lower relative to historical comparisons (adjRR 0.82 to 0.85). For AL residents, the hospitalization rate

**Table 1. Yearly characteristics of Alberta Assisted Living (AL) and Nursing Home (NH) residents,[a] comparing select time periods.**

| Characteristic | March 2018 | | March 2019 | | March 2020 | | March 2021 | | Dec 2021 | |
|---|---|---|---|---|---|---|---|---|---|---|
| | AL (n = 8431) | NH (n = 13695) | AL (n = 9069) | NH (n = 14191) | AL (n = 9485) | NH (n = 14319) | AL (n = 9051) | NH (n = 13396) | AL (n = 9323) | NH (n = 14155) |
| Age, mean (SD), y | 80.4(12.8)[b] | 81.9 (12.5) | 80.7 (12.5) | 81.7 (12.6) | 80.7 (12.4) | 81.4 (12.9) | 80.5 (12.5) | 81.1 (12.9) | 80.5 (12.5) | 81.1 (12.8) |
| Age group, y | | | | | | | | | | |
| <65 | 1060 (12.6) | 1395 (10.2) | 1054 (11.6) | 1478 (10.4) | 1079 (11.4) | 1535 (10.7) | 1100 (12.2) | 1497 (11.2) | 1101 (11.8) | 1535 (10.8) |
| 65–74 | 1077 (12.8) | 1674 (12.2) | 1161 (12.8) | 1873 (13.2) | 1298 (13.7) | 1975 (13.8) | 1261 (13.9) | 1957 (14.6) | 1312 (14.1) | 2106 (14.9) |
| 75–84 | 2312 (27.4) | 3645 (26.6) | 2603 (28.7) | 3753 (26.5) | 2731 (28.8) | 3761 (26.3) | 2562 (28.3) | 3537 (26.4) | 2695 (28.9) | 3751 (26.5) |
| 85+ | 3982 (47.2) | 6981 (51.0) | 4251 (46.9) | 7087 (49.9) | 4377 (46.2) | 7048 (49.2) | 4128 (45.6) | 6405 (47.8) | 4215 (45.2) | 6763 (47.8) |
| Female | 5664 (67.2) | 8767 (64.0) | 6020 (66.4) | 8958 (63.1) | 6330 (66.7) | 8995 (62.8) | 6075 (67.1) | 8408 (62.8) | 6208 (66.6) | 8867 (62.6) |
| ADL-H 3+ | | | | | | | | | | |
| Extensive assist/ Dependent | 2219 (26.3)[b] | 11355 (82.9) | 2329 (25.7)[b] | 11824 (83.3) | 2430 (25.6)[b] | 12034 (84.0) | 2387 (26.4)[b] | 11325 (84.5) | 2465 (26.4)[b] | 11924 (84.2) |
| CPS 3+ | | | | | | | | | | |
| Moderate-severe impairment | 2488 (29.5)[b] | 8554 (62.5) | 2647 (29.2)[b] | 8740 (61.6) | 2670 (28.2)[b] | 8717 (60.9) | 2509 (27.7)[b] | 8154 (60.9) | 2664 (28.6)[b] | 8580 (60.6) |
| DRS 3+ | | | | | | | | | | |
| Depressive symptoms | 1532 (18.2)[b] | 4073 (29.7) | 1686 (18.6)[b] | 3997 (28.2) | 1634 (17.2)[b] | 3907 (27.3) | 1431 (15.8)[b] | 3523 (26.3) | 1338 (14.4)[b,c] | 3675 (26.0) |
| CHESS Scale 2+ | | | | | | | | | | |
| Higher health instability | 986 (11.7)[b] | 2955 (21.6) | 1074 (11.8)[b] | 2968 (20.9) | 1060 (11.2)[b] | 2975 (20.8) | 930 (10.3)[b] | 2809 (21.0) | 1019 (10.9)[b] | 2968 (21.0) |
| Frailty (FI value) | | | | | | | | | | |
| Not frail (<0.2) | 2977 (35.3)[b] | 1404 (10.3) | 3175 (35.0)[b] | 1598 (11.3) | 3375 (35.6)[b] | 1725 (12.1) | 3133 (34.6)[b] | 1721 (12.9) | 3271 (35.1)[b] | 1960 (13.9) |
| Prefrail (0.2–0.3) | 3065 (36.4)[b] | 3985 (29.1) | 3446 (38.0)[b] | 4234 (29.8) | 3604 (38.0)[b] | 4470 (31.2) | 3425 (37.8)[b] | 4320 (32.3) | 3509 (37.6) | 4732 (33.4) |
| Frail (>0.3) | 2389 (28.3)[b] | 8305 (60.6) | 2448 (27.0)[b] | 8359 (58.9) | 2506 (26.4)[b] | 8124 (56.7) | 2493 (27.5)[b] | 7355 (54.9) | 2543 (27.3)[b] | 7456 (52.7)[c] |
| # Responsive behaviours[d] | | | | | | | | | | |
| 2+ | 787 (9.3)[b] | 3194 (23.3) | 820 (9.0)[b] | 3300 (23.3) | 810 (8.5)[b] | 3281 (22.9) | 741 (8.2)[b] | 3153 (23.5) | 739 (7.9)[b] | 3284 (23.2) |
| Chronic Conditions | | | | | | | | | | |
| mean (SD) | 3.77 (1.89) | 3.87 (1.88) | 3.83 (1.91) | 3.84 (1.88) | 3.85 (1.89) | 3.82 (1.89) | 3.85 (1.89) | 3.71 (1.89) | 3.84 (1.9)[b] | 3.56 (1.86)[c] |
| # | | | | | | | | | | |
| 0–2 | 2206 (26.2) | 3309 (24.2) | 2278 (25.1) | 3491 (24.6) | 2323 (24.5) | 3613 (25.2) | 2225 (24.6) | 3703 (27.6) | 2304 (24.7)[b] | 4263 (30.1)[c] |
| 3,4 | 3490 (41.4) | 5698 (41.6) | 3720 (41.0) | 5893 (41.5) | 3901 (41.1) | 5868 (41.0) | 3744 (41.4) | 5452 (40.7) | 3828 (41.1) | 5812 (41.1) |
| 5+ | 2735 (32.4) | 4688 (34.2) | 3071 (33.9) | 4807 (33.9) | 3261 (34.4) | 4838 (33.8) | 3082 (34.1) | 4241 (31.7) | 3191 (34.2)[b] | 4080 (28.8)[c] |
| Selected Health Conditions | | | | | | | | | | |
| Dementia | 4781 (56.7)[b] | 8757 (63.9) | 5232 (57.7)[b] | 8937 (63.0) | 5500 (58.0) | 8830 (61.7) | 5277 (58.3) | 8012 (59.8) | 5439 (58.3) | 8225 (58.1)[c] |
| Arthritis | 3899 (46.3) | 5815 (42.5) | 4205 (46.4) | 6048 (42.6) | 4387 (46.3) | 6015 (42.0) | 4241 (46.9)[b] | 5320 (39.7) | 4311 (46.2)[b] | 5247 (37.1)[c] |
| Osteoporosis | 2174 (25.8) | 4008 (29.3) | 2300 (25.4) | 3982 (28.1) | 2384 (25.1) | 3991 (27.9) | 2215 (24.5) | 3511 (26.2) | 2332 (25.0) | 3457 (24.4) |
| Diabetes | 1937 (23.0) | 3586 (26.2) | 2119 (23.4) | 3780 (26.6) | 2211 (23.3) | 3826 (26.7) | 2110 (23.3) | 3575 (26.7) | 2177 (23.4) | 3720 (26.3) |
| COPD | 1721 (20.4) | 2807 (20.5) | 1883 (20.8) | 2824 (19.9) | 1951 (20.6) | 2734 (19.1) | 1801 (19.9) | 2489 (18.6) | 1828 (19.6) | 2357 (16.7) |
| Congestive Heart Failure | 980 (11.6) | 2028 (14.8) | 1055 (11.6) | 2032 (14.3) | 1080 (11.4) | 1941 (13.6) | 1029 (11.4) | 1693 (12.6) | 1034 (11.1) | 1670 (11.8) |

(*Continued*)

**Table 1.** (Continued)

| Characteristic | March 2018 | | March 2019 | | March 2020 | | March 2021 | | Dec 2021 | |
|---|---|---|---|---|---|---|---|---|---|---|
| | AL (n = 8431) | NH (n = 13695) | AL (n = 9069) | NH (n = 14191) | AL (n = 9485) | NH (n = 14319) | AL (n = 9051) | NH (n = 13396) | AL (n = 9323) | NH (n = 14155) |
| Cancer | 535 (6.4)[b] | 1487 (10.9) | 609 (6.7)[b] | 1582 (11.2) | 620 (6.5)[b] | 1612 (11.3) | 571 (6.3)[b] | 1474 (11.0) | 615 (6.6)[b] | 1506 (10.6) |

Abbreviations: ADL-H, activities of daily living self-performance hierarchy scale; CHESS, Changes in Health, End-Stage Disease, Signs & Symptoms; COPD, chronic obstructive pulmonary disease; CPS, Cognitive Performance Scale; DRS, Depression Rating Scale; FI, frailty index; SD, standard deviation.

a Unless indicated otherwise, data are expressed as Column No. (%) of residents with percentages rounded.

b Standardized difference of >0.10 considered clinically meaningful difference (comparing AL to NH)

c Standardized difference of >0.10 considered clinically meaningful difference (comparing Dec 2021 to Mar 2019)

d Sum of 4 behaviours assessed on the RAI-HC (verbal abuse, physical abuse, socially inappropriate/disruptive care, resisting care)

increased after the initial wave 1 decline and showed no statistically significant difference relative to historical periods during the peak of wave 2 (December 2020) and the subsequent month (January 2021). During January 2021, the hospitalization rate among AL residents increased by 27% relative to that among NH residents (period*setting interaction p = 0.013).

**Table 2. Yearly home characteristics of Alberta Assisted Living (AL) and Nursing Home (NH) residents,[a] comparing select time periods.**

| Home Characteristic | March 2018 | | March 2019 | | March 2020 | | March 2021 | | Dec 2021 | |
|---|---|---|---|---|---|---|---|---|---|---|
| | AL (n = 8431) | NH (n = 13695) | AL (n = 9069) | NH (n = 14191) | AL (n = 9485) | NH (n = 14319) | AL (n = 9051) | NH (n = 13396) | AL (n = 9323) | NH (n = 14155) |
| **Geographic Health Zone** | | | | | | | | | | |
| Calgary | 2460 (29.2)[b] | 5005 (36.6) | 2607 (28.8)[b] | 5363 (37.8) | 2882 (30.4)[b] | 5480 (38.3) | 2717 (30.0)[b] | 5181 (38.7) | 2777 (29.8)[b] | 5701 (40.3) |
| Central | 1216 (14.4) | 2088 (15.3) | 1351 (14.9) | 2123 (15.0) | 1418 (15.0) | 2117 (14.8) | 1494 (16.5) | 2072 (15.5) | 1555 (16.7) | 2069 (14.6) |
| Edmonton | 2843 (33.7) | 4663 (34.1) | 2830 (31.2) | 4774 (33.6) | 2929 (30.9) | 4842 (33.8) | 2661 (29.4) | 4277 (31.9) | 2811 (30.2) | 4606 (32.5) |
| North | 630 (7.5) | 1160 (8.5) | 754 (8.3) | 1124 (7.9) | 760 (8.0) | 1107 (7.7) | 742 (8.2) | 1086 (8.1) | 800 (8.6) | 1024 (7.2) |
| South | 1282 (15.2)[b] | 779 (5.7) | 1527 (16.8)[b] | 807 (5.7) | 1496 (15.8)[b] | 773 (5.4) | 1437 (15.9)[b] | 780 (5.8) | 1380 (14.8)[b] | 755 (5.3) |
| **Location** | | | | | | | | | | |
| Rural | 848 (10.1)[b] | 1844 (13.5) | 981 (10.8) | 1769 (12.5) | 953 (10.1) | 1732 (12.1) | 1003 (11.1) | 1747 (13.0) | 1012 (10.9) | 1642 (11.6) |
| Urban | 7583 (89.9)[b] | 11851 (86.5) | 8088 (89.2) | 12422 (87.5) | 8532 (90.0) | 12587 (87.9) | 8048 (88.9) | 11649 (87.0) | 8311 (89.2) | 12513 (88.4) |
| Bed Size, median (25th, 75th) | 86 (45,107) | 129 (73,220) | 85 (45,107) | 132 (74,210) | 84 (48,107) | 132 (74,210) | 84 (46,107) | 127 (69,208) | 84 (48,107) | 132 (74,208) |
| **Bed Size (median cut-point)** | | | | | | | | | | |
| ≤102 | 5830 (69.2)[b] | 5361 (39.2) | 6258 (69.0)[b] | 5542 (39.1) | 6559 (69.2)[b] | 5627 (39.3) | 6318 (69.8)[b] | 5453 (40.7) | 6509 (69.8)[b] | 5506 (38.9) |
| >102 | 2592 (30.7)[b] | 8334 (60.9) | 2805 (30.9)[b] | 8649 (61.0) | 2926 (30.9)[b] | 8692 (60.7) | 2733 (30.2)[b] | 7943 (59.3) | 2814 (30.2)[b] | 8649 (61.1) |
| **Ownership Status** | | | | | | | | | | |
| AB Health Services (gov't) | 393 (4.7)[b] | 4323 (31.6) | 434 (4.8)[b] | 4276 (30.1) | 435 (4.6)[b] | 4277 (29.9) | 480 (5.3)[b] | 4074 (30.4) | 499 (5.4)[b] | 4126 (29.2) |
| Non-Profit / Voluntary | 3651 (43.3)[b] | 4516 (33.0) | 3846 (42.4)[b] | 4580 (32.3) | 3973 (41.9)[b] | 4624 (32.3) | 3748 (41.4)[b] | 4344 (32.4) | 3770 (40.4)[b] | 4596 (32.5) |
| For-Profit / Private | 4387 (52.0)[b] | 4856 (35.5) | 4789 (52.8)[b] | 5335 (37.6) | 5077 (53.5)[b] | 5418 (37.8) | 4823 (53.3)[b] | 4978 (37.2) | 5054 (54.2)[b] | 5433 (38.4) |

a Unless indicated otherwise, data are expressed as Column No. (%) of residents with percentages rounded.

b Standardized difference of >0.10 considered clinically meaningful difference (comparing AL to NH)

Both settings showed significantly lower hospitalization rates during pandemic waves 3 (April 2021) and 4 (September 2021), with a more pronounced decrease during wave 4 (AL: adjRR 0.76, 95% CI 0.66–0.88; NH: adjRR 0.65, 95% CI 0.56–0.75).

Among AL residents, there were no statistically significant associations between the *selected* pandemic periods and rate of hospitalization with a delayed discharge (Table 3, Fig 2 and S2 Fig), whereas among NH residents, there was a statistically significant increase during both the peak of wave 1 (adjRR 1.94, 95% CI 1.31–2.85, with a statistically significant period*setting interaction, p = 0.008) and August 2020 (adjRR 1.84, 95% CI 1.19–2.86). Of note, both settings showed a statistically significant increase in rate of hospitalization with a delayed discharge during October 2020 (not an a priori exposure period of interest). This increase was more pronounced for NH (adjRR 2.57 (1.69–3.91) than AL (adjRR 1.55 (1.17–2.04) residents (period*-setting interaction, p = 0.048).

Among AL residents, there was a statistically significant increase in rate of hospitalization with death during the peak of wave 2 and in January 2021, followed by a significant decrease in this outcome during the peak of wave 3 (Table 3, S3 Fig). For NH residents, the only statistically significant association was for an increase in rate of hospitalization with death during wave 2 relative to historical period.

Both sensitivity analyses produced comparable findings to those presented in Table 3.

Rates of all-cause hospitalization and hospitalization with a delayed discharge were consistently higher among AL than NH residents throughout the study period (adjRR of 1.4–1.6 and

**Table 3. Adjusted rate ratios[a] for all-cause hospitalization, hospitalization with delayed discharge, and hospitalization with death, comparing COVID-19 pandemic vs historical (2018/19) monthly periods, among Alberta Assisted Living (AL) and Nursing Home (NH) residents.**

| | Rate Ratio (95% CI) for Hospital Outcome Associated with Select COVID-19 Pandemic Time Periods | | | | | |
| --- | --- | --- | --- | --- | --- | --- |
| | April 2020 | August 2020 | December 2020 | January 2021 | April 2021 | September 2021 |
| **Hospital Outcome & Setting** | [Peak Wave 1] | | [Peak Wave 2] | | [Peak Wave 3] | [Peak Wave 4] |
| *All-Cause Hospitalization* | | | | | | |
| Period (pandemic vs 2018/19) | | | | | | |
| AL | **0.60 (0.51–0.71)** | **0.87 (0.76–1.00)** | 0.92 (0.80–1.04) | 1.05 (0.92–1.18)[c] | **0.84 (0.73–0.97)** | **0.76 (0.66–0.88)** |
| NH | **0.74 (0.64–0.85)** | **0.82 (0.71–0.95)** | **0.82 (0.72–0.95)** | **0.82 (0.72–0.95)[c]** | **0.85 (0.74–0.98)** | **0.65 (0.56–0.75)** |
| *Urgent Hospitalization* | | | | | | |
| Period (pandemic vs 2018/19) | | | | | | |
| AL | **0.62 (0.52–0.73)** | 0.89 (0.77–1.02) | 0.93 (0.81–1.06) | 1.05 (0.93–1.19)[c] | **0.83 (0.72–0.97)** | **0.76 (0.65–0.88)** |
| NH | **0.71 (0.61–0.83)** | **0.83 (0.72–0.96)** | **0.82 (0.71–0.95)** | **0.83 (0.72–0.95)[c]** | **0.86 (0.74–0.99)** | **0.67 (0.57–0.78)** |
| *Hospitalization with Delayed Discharge[b]* | | | | | | |
| Period (pandemic vs 2018/19) | | | | | | |
| AL | 0.98 (0.71–1.34)[d] | 1.14 (0.87–1.51) | 0.98 (0.73–1.33) | 1.16 (0.88–1.52) | 0.81 (0.57–1.15) | 0.71 (0.50–1.01) |
| NH | **1.94 (1.31–2.85)[d]** | **1.84 (1.19–2.86)** | 1.39 (0.89–2.15) | 1.15 (0.73–1.80) | 0.82 (0.49–1.38) | 0.63 (0.35–1.15) |
| *Hospitalization with Death* | | | | | | |
| Period (pandemic vs 2018/19) | | | | | | |
| AL | 0.80 (0.55–1.17) | 1.27 (0.88–1.82) | **1.89 (1.40–2.54)** | **1.47 (1.07–2.01)** | **0.50 (0.31–0.79)** | 1.20 (0.77–1.87) |
| NH | 1.05 (0.75–1.46) | 0.81 (0.57–1.14) | **1.53 (1.14–2.05)** | 0.99 (0.72–1.37) | 0.71 (0.48–1.05) | 0.90 (0.63–1.29) |

a For each time period, separate Poisson regression generalized estimating equations (GEE) models were used to estimate rate ratios for period (COVID-19 pandemic vs historical [2018/19]), setting (AL vs NH) and period-setting interactions; Models adjusted for age, sex, ADL, CPS, CHESS, #chronic conditions, health zone & ownership status. See S2 Table for all model estimates, including setting.

b Note: An adjusted model for October 2020 was also computed given the rise in rate of hospitalization with a delayed discharge evident in Fig 2 and S2 Fig: for AL the adjusted RR = 1.55 (1.17–2.04), for NH the adjusted RR = 2.57 (1.69–3.91), with a statistically significant interaction (period*setting), p = 0.048

c Test of statistical significance for interaction of period*setting, p = 0.013

d Test of statistical significance for interaction of period*setting, p = 0.008

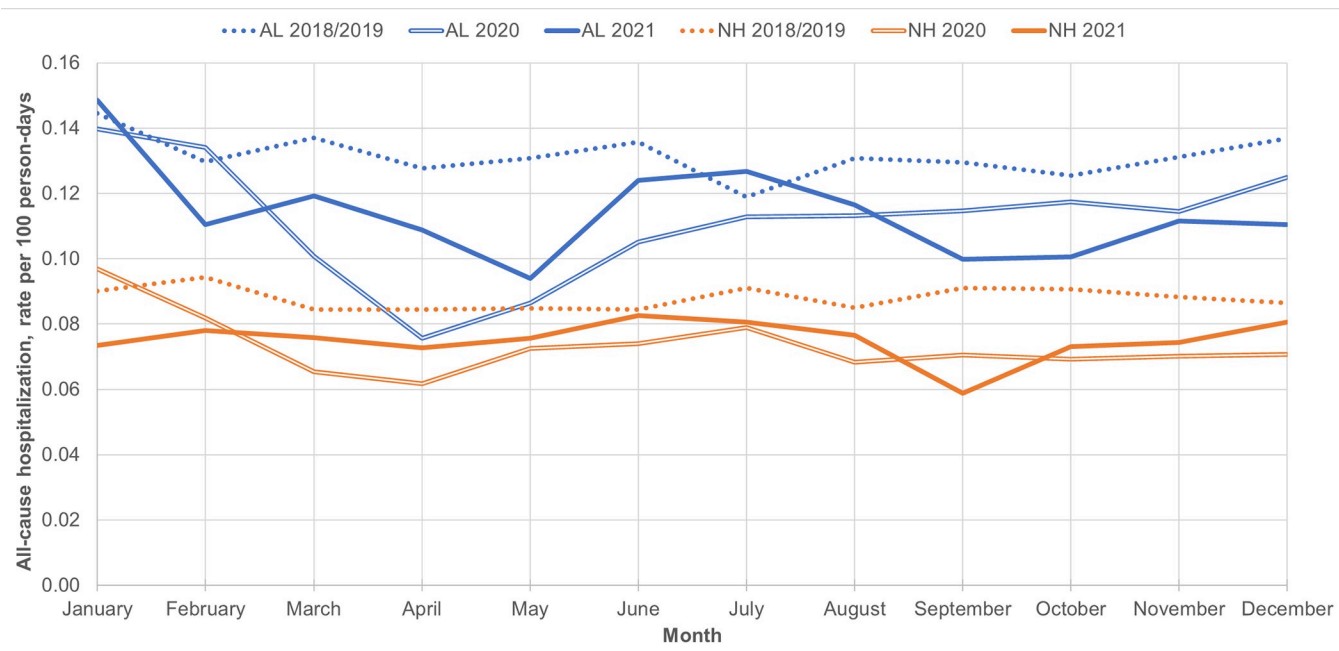

**Fig 1. Monthly all-cause hospitalization rate (per 100 person-days) for historical (2018/19 combined) and 2020–2021 pandemic periods, among Assisted Living (AL) and Nursing Home (NH) residents.**

4.5–6.5, respectively; **Figs 1 and 2, S2 Table**). AL residents consistently showed higher median delayed discharge bed days (e.g., during January 2021, median 30 [IQR 6–52] bed days for AL vs median 9.5 [IQR 4–28] for NH residents, **S3 Table**).

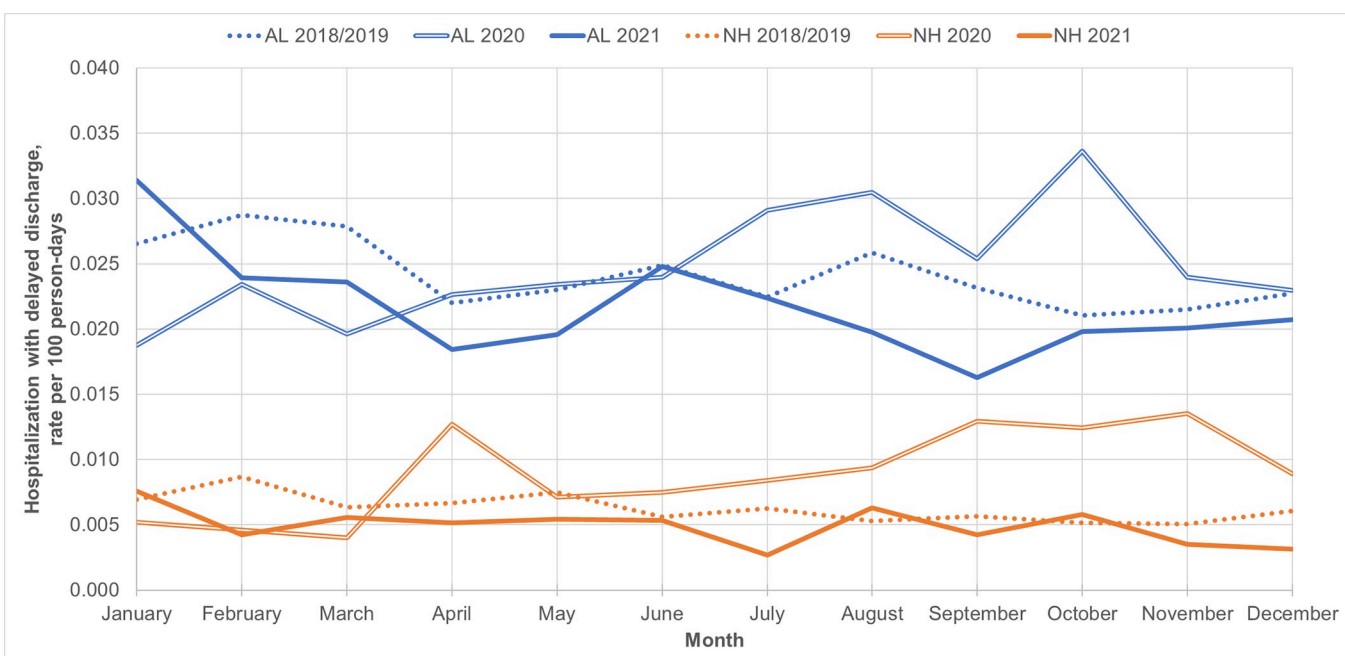

**Fig 2. Monthly hospitalization with delayed discharge rate (per 100 person-days) for historical (2018/19 combined) and 2020–2021 pandemic periods, among Assisted Living (AL) and Nursing Home (NH) residents.**

Among the top 10 ICD-10-CA codes for the most responsible diagnosis (for hospitalizations during April 1, 2020 and January 1, 2021 vs historical counterparts) (**S3 Table**), dementia/mental health/delirium and injuries/fractures were more common and ranked higher among AL vs NH residents (especially during the pandemic months), whereas sepsis and renal failure/UTIs were more common and ranked higher during pandemic months among NH residents. Respiratory diseases were among the top two diagnoses for both settings but accounted for smaller proportion of diagnoses during pandemic periods (replaced by COVID-19 as the top condition in January 2021).

The monthly rates of COVID-19 hospitalizations were highest during wave 2, decreased following the implementation of vaccinations, and increased again during wave 4 (**S4 Fig**). During waves 2 and 4 the rates were higher among AL than NH residents. The proportion of all hospitalizations that were for COVID-19 was comparable for AL and NHs during December 2020 (27.0% vs 26.1%) and slightly higher for AL in January 2021 (19.5% vs 14.9%) and September 2021 (7.7% vs 5.6%). Monthly rates of positive COVID-19 tests were similar in AL and NHs (except for wave 1) and aligned with peak surges observed for COVID-19 hospitalizations (**S5 Fig**).

## Discussion

In this population-based study of residents of AL and NHs in Alberta Canada, both settings showed pronounced decreases in all-cause hospitalizations during the first (40% and 26%, respectively) and fourth (24% and 35%, respectively) COVID-19 pandemic peaks but they varied significantly in this outcome during wave 2, and in the secondary hospital outcomes examined during selected pandemic periods. Whereas hospitalization rates among NH residents remained significantly below historical levels throughout, they increased for AL residents (by 27% relative to NHs) and were not significantly different from pre-pandemic rates during this period. For hospitalization with a delayed discharge, rates increased significantly during wave 1 (and into August 2020) for NH residents only, and though both settings showed a significant increase in rates of hospitalization with death during wave 2, this increase was larger and persisted longer (beyond peak of wave 2) for AL residents. The two settings also showed differences in the distribution of most responsible hospital diagnoses during pandemic periods and in COVID-19 associated hospitalizations (higher in AL than NH for waves 2 and 4), the latter despite generally comparable rates of positive COVID-19 tests.

The substantial decline in hospitalizations noted for both settings during the peak of wave 1 (consistent for all-cause and emergent admissions) aligns with other studies of chronically ill older adults in community [21] and long-term care settings [23–25, 28]. Few studies of congregate care have examined changes in healthcare use for AL residents [24] or for periods beyond 2020 (following the introduction of COVID-19 vaccinations). Expanding on earlier work, our findings reveal that with one exception (wave 2), pandemic periods examined up to the end of 2021 were associated with significantly lower hospitalizations from both settings. Complex and inter-related health system, home and resident/family factors may explain these lower rates, including the protection of acute care beds for the influx of expected COVID-19 cases early on in the pandemic [21], a reluctance by hospitals to accept AL or NH residents and/or by care homes or family to transfer residents to hospital because of the perceived risk of infection [25], changes in resident conditions typically associated with hospitalization [23, 24, 27], and a failure to recognize emergent health conditions requiring acute care given staffing shortages and family absence [27]. From a resident quality of care perspective, these findings may be viewed positively (e.g., there were fewer unnecessary hospital transfers from care home settings) [18, 43, 44] or negatively (e.g., there were more residents with serious unmet clinical

care needs at heightened risk for death or functional and cognitive decline) [19, 24, 45]. Both possibilities warrant further investigation.

The evident rise in hospital admissions for AL residents in wave 2 (back to historical levels), may reflect an increased likelihood for AL homes to transfer residents with COVID-19 and other conditions to hospital and/or greater illness severity in AL vs NH residents (perhaps facilitated by less effective isolation measures or greater interactions among AL residents) during this period [20]. In both settings, positive COVID-19 cases among residents were especially high during wave 2, dropped dramatically following widespread resident vaccinations, and increased again during wave 4. It may also be that AL residents experienced more adverse health outcomes (e.g., inappropriate medication use [46], falls/injuries [47], worse mental health and/or disease progression) and potentially avoidable hospitalizations during this surge because of reductions in already low levels of professional staffing and services [8–10, 12–14] and absence of family caregivers [3]. Our data regarding setting-specific changes in the distribution of most responsible diagnoses for hospitalizations during waves 1 and 2 and consistently elevated rates of hospitalization (including admissions with delayed discharge and higher bed stays) for AL relative to NHs throughout pre- and post-pandemic periods partially support this explanation.

Though there is no evidence of changes in AL or NH admission and discharge criteria during the pandemic [48], there may have been differences in the enactment of these criteria across care settings and/or adoption of strategies to prevent avoidable hospitalizations (e.g., increased communication between NH sites and emergency department physicians or involvement of community paramedics, [49]) that contributed to the setting-specific hospitalization findings noted above.

The increased rate ratios for hospital admission with a delayed discharge shown for NH residents early in the pandemic, and for both settings during Oct 2020, may be related to the reduced availability of NH and AL beds as homes addressed concerns of overcrowding and the need for isolation spaces during periods of peak COVID-19 cases [21, 50]. These findings may also reflect hesitation among physicians, staff and/or families to have residents discharged back to AL or NHs during peak surges because of a limited availability of staff to care for residents who likely had increased health needs.

Both settings experienced a significant increase in hospitalization with death during the peak of wave 2 relative to historical periods (coinciding with peak COVID-19 cases and hospitalizations), but the magnitude of this increase was larger and persisted into early 2021 for AL residents. As suggested by our data on most responsible diagnoses, AL residents were more likely to be admitted with health conditions associated with increased hospital bed days and mortality (e.g., dementia, mental health issues, delirium) [51]. They may also have presented with more severe health conditions related to less timely detection of emerging health issues and limited ability to augment care in the AL home [8, 13]. These hospital deaths are disconcerting given residents' and families' preferred place of death (at home) and that many residents likely died alone because of public health restrictions.

## Study strengths & limitations

Strengths of this study include its use of population-based clinical and health administrative data and rigorous methodology (including adjustment for relevant confounding factors), comparison of hospitalization trends for residents of AL (a less researched setting) and NHs, and examination of outcomes beyond initial pandemic waves and for periods following widespread COVID-19 vaccinations among residents. Limitations include the lack of data on setting-specific differences in staffing, service availability, advance directives (or residents' goals of care),

policy changes and resident health outcomes during the pandemic waves examined. Though our models adjusted for key covariates, there may have been temporary disruptions in completing RAI assessments early in the pandemic. We cannot comment on the appropriateness of the hospitalization trends observed and future investigations of associated outcomes for residents and families are warranted. We did not have access to other data of potential relevance to hospitalization events observed across pandemic waves, including the variants of SARS-CoV-2 present throughout the study period. As our study was restricted to publicly subsidized AL in Alberta, caution is needed in generalizing our findings to other AL settings in Canada and elsewhere.

## Conclusions

Hospitalizations represent an important quality of care indicator for AL and NH populations, reflecting the importance of potentially preventable hospital transfers as well as equitable access to appropriate and timely medical care. This study showed that the associations between various pandemic waves and hospitalization events varied in important ways for AL compared with NH residents, further highlighting the unique challenges faced by different congregate care settings during COVID-19 [4, 5] and their varying responses to this external stress. Our findings support the presence of setting-specific drivers of residents' healthcare use and outcomes both prior to and during COVID-19 that require careful consideration by continuing care operators, health care providers and policy decision makers. These setting-specific differences should also inform future interventions [44, 52, 53] aimed at strengthening the integration, communication and support between congregate care and acute care settings. Considerations of the clinical complexity of AL residents and hospitalization trends observed (both historically and during COVID-19) strongly support previous calls for a blended social-medical model of care in AL homes [1, 6, 16]. Strategies to be investigated include more routine access to primary care physicians and/or nurse practitioners (both on-site and via telemedicine) [54], enhanced levels of well-trained professional staff, clearly documented advance directives, and the development of policies and procedures specific to the appropriate detection, treatment, and management of the chronic physical and mental health conditions common in AL residents.

## Supporting information

**S1 Table. Description of Alberta provincial clinical and health administrative databases.**
(DOCX)

**S2 Table. Adjusted rate ratios for all-cause hospitalization, hospitalization with delayed discharge, and hospitalization with death, comparing COVID-19 pandemic vs historical (2018/19) monthly periods, among Alberta Assisted Living (AL) and Nursing Home (NH) residents.**
(DOCX)

**S3 Table. A) Distribution of top 10 ICD-10-CA diagnosis (grouped) for most responsible diagnosis associated with hospitalizations during April 2020 and January 2021 (relative to comparable historical 2018/19 months), by setting.** B) Distribution of top 10 ICD-10-CA diagnosis / chapters for most responsible diagnosis associated with hospitalizations during April 2020 and January 2021 (relative to comparable historical 2018/19 months), by setting.
(DOCX)

**S1 Fig. Monthly all-cause hospitalization rate (per 100 person-days) across study period, January 1, 2018 to December 31, 2021, among Assisted Living (AL) and Nursing Home**

(NH) residents.
(DOCX)

**S2 Fig. Monthly hospitalization with delayed discharge rate (per 100 person-days) across study period, January 1, 2018 to December 31, 2021, among Assisted Living (AL) and Nursing Home (NH) residents.**
(DOCX)

**S3 Fig. Monthly hospitalization with death rate (per 100 person-days) across study period, January 1, 2018 to December 31, 2021, among Assisted Living (AL) and Nursing Home (NH) residents.**
(DOCX)

**S4 Fig. Monthly COVID-19 hospitalization rate (per 100 person-days) across pandemic period, March 1, 2020 to December 31, 2021, among Assisted Living (AL) and Nursing Home (NH) residents.**
(DOCX)

**S5 Fig. Monthly rate (per 100 residents) of positive COVID-19 tests, across pandemic period, March 1, 2020 to December 31, 2021, among Assisted Living (AL) and Nursing Home (NH) residents.**
(DOCX)

**S1 Text. RECORD statement.**
(PDF)

## Acknowledgments

We would like to thank the Alberta SPOR SUPPORT Unit for their assistance in accessing the data used in this study.

## Author Contributions

**Conceptualization:** Colleen J. Maxwell, Eric McArthur, David B. Hogan, Hana Dampf, Jeffrey Poss, Joseph E. Amuah, Susan E. Bronskill, Matthias Hoben.

**Data curation:** Colleen J. Maxwell, Erik Youngson, Zoe Hsu, Matthias Hoben.

**Formal analysis:** Eric McArthur, Hana Dampf.

**Funding acquisition:** Colleen J. Maxwell, Matthias Hoben.

**Investigation:** Colleen J. Maxwell, Eric McArthur, Hana Dampf.

**Methodology:** Colleen J. Maxwell, Eric McArthur, Hana Dampf, Jeffrey Poss, Matthias Hoben.

**Project administration:** Colleen J. Maxwell, Matthias Hoben.

**Resources:** Erik Youngson, Zoe Hsu, Matthias Hoben.

**Software:** Erik Youngson, Zoe Hsu.

**Supervision:** Colleen J. Maxwell, Matthias Hoben.

**Visualization:** Colleen J. Maxwell, Eric McArthur, Hana Dampf.

**Writing – original draft:** Colleen J. Maxwell.

**Writing – review & editing:** Eric McArthur, David B. Hogan, Hana Dampf, Jeffrey Poss, Joseph E. Amuah, Susan E. Bronskill, Erik Youngson, Zoe Hsu, Matthias Hoben.

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
