## [Decision Letter · Decision Letter 0]

7 May 2024

PONE-D-24-03317Comparison of hospitalization events among residents of assisted living and nursing homes during COVID-19: do settings respond differently during public health crises?PLOS ONE

Dear Dr. Maxwell,

Thank you for submitting your manuscript to PLOS ONE. After careful consideration, we feel that it has merit but does not fully meet PLOS ONE’s publication criteria as it currently stands. Therefore, we invite you to submit a revised version of the manuscript that addresses the points raised during the review process.

We look forward to receiving your revised manuscript.

Kind regards,

Mario Ulises Pérez-Zepeda, M.D., Ph.D.

Academic Editor

PLOS ONE

Journal Requirements:

Reviewers' comments:

Reviewer's Responses to Questions

**Comments to the Author**

1. Is the manuscript technically sound, and do the data support the conclusions?

Reviewer #1: Yes

Reviewer #2: Yes

2. Has the statistical analysis been performed appropriately and rigorously? 

Reviewer #1: Yes

Reviewer #2: Yes

3. Have the authors made all data underlying the findings in their manuscript fully available?

Reviewer #1: Yes

Reviewer #2: Yes

4. Is the manuscript presented in an intelligible fashion and written in standard English?

Reviewer #1: Yes

Reviewer #2: Yes

5. Review Comments to the Author

Reviewer #1: Well written.

The practical problems of AL have been brought out in the first paragraph itself.

The residents of the NH were frailer, had more comorbidities, cognitively worse and were dependent on ADLs, as expected. Did this approach statistical significance?

Among AL residents, there was a statistically significant increase in rate of hospitalization with death during the peak of wave 2 and in January 2021, followed by a significant decrease in this outcome during the peak of wave 3. For NH residents, the only statistically significant association was for an increase in rate of hospitalization with death during wave 2 relative to historical period. This is true all over the world. Wave 2 was the worst and by wave 3 and 4 – vaccinations had been given, and others would have been exposed to the virus. But what about variants?

During waves 2 and 4 the rates were higher among AL than NH residents- this could be attributed to the fact that AL patients mingled, while NH patients were bedbound and exposed to fewer people – both inmates and carers at the facilities.

Did death and burden of comorbidities correlate? What about morbidity?

Did the admission and discharge criteria vary between waves, and between NH and AL?

Reviewer #2: The article is well constructed and gives an interesting perspective of how different settings of assisted living and nursing homes respond during health crises. Some data should be clarified and be less dispersive to ensure that readers undestand exactly the researches point of view. I did appreciate the strategies to be investigated in the conclusions and I think it worths finding out more about.

6. PLOS authors have the option to publish the peer review history of their article (what does this mean?). If published, this will include your full peer review and any attached files.

Reviewer #1: No

Reviewer #2: No

---

## [Author Response · Author response to Decision Letter 0]

14 May 2024

Response to Reviewers: Note: Comments from Reviewers are presented verbatim using italics. The location of revised text refers to lines in the revised manuscript with track changes.

Reviewer #1:

1. Well written. The practical problems of AL have been brought out in the first paragraph itself.

Response: Thank you for noting these strengths of our manuscript.

2. The residents of the NH were frailer, had more comorbidities, cognitively worse and were dependent on ADLs, as expected. Did this approach statistical significance?

Response: In our comparison of the characteristics of residents in AL vs NH, we reported which differences were larger than a standardized difference of 0.10 as a measure of a meaningful difference (please see footnotes used in Tables 1 and 2). We used standardized differences rather than p values given the large sample sizes involved - which would result in small differences being statistically significant but not necessarily clinically relevant. In our original manuscript, we noted the rationale for this in the Analysis section (lines 172-174) with a supporting reference (reference #42).

3. Among AL residents, there was a statistically significant increase in rate of hospitalization with death during the peak of wave 2 and in January 2021, followed by a significant decrease in this outcome during the peak of wave 3. For NH residents, the only statistically significant association was for an increase in rate of hospitalization with death during wave 2 relative to historical period. This is true all over the world. Wave 2 was the worst and by wave 3 and 4 – vaccinations had been given, and others would have been exposed to the virus. But what about variants?

Response: We do acknowledge that COVID-19 case rates were most pronounced (in Alberta and globally) during the period of the pandemic prior to widespread vaccinations. This is also evident for residents in our study as illustrated in our S5 Fig. Regarding our findings for rate of hospitalization with death – a key point we wished to highlight with these statements were the differences in this outcome observed between AL and NH beyond the peak of wave 2 – i.e., during January 2021 and during the peak of wave 3 (as also emphasized in our Discussion – see lines 305-307 and lines 356-359). Related to this was our finding of considerably higher COVID-19 hospitalization rates among residents of AL vs NH during the peak of wave 2 and in the subsequent month of January 2021 (as shown in S4 Fig.). As noted throughout our manuscript, there were several notable differences in study outcomes between those in AL vs NH – including during wave 2 of the COVID-19 pandemic.

We are not able to comment on the relevance of differences in specific variants of SARS-CoV-2 present throughout the study period as we did not have access to these data (we have added this statement to the limitation section of our Discussion, see lines 377-379). We do not believe that the key differences we observed in outcomes between AL and NH settings would be due to differences in the variants present, as the dominant variants would be expected to be comparable across settings at specific points in time.

4. During waves 2 and 4 the rates were higher among AL than NH residents- this could be attributed to the fact that AL patients mingled, while NH patients were bedbound and exposed to fewer people – both inmates and carers at the facilities.

Response: Thank you for this comment regarding potential contributing factors to the observed differences between AL and NH settings in the monthly rates of COVID-19 hospitalizations. As we noted in our Results (lines 290-295) and Discussion (lines 309-310) sections, these differences were observed during waves 2 and 4 even though monthly rates of positive COVID-19 tests were relatively similar in ALs and NHs during these waves (as illustrated in our S5 Fig.). We do comment on possible reasons for these differences in our Discussion (including an increased likelihood for AL homes to transfer residents with COVID-19 and/or greater illness severity in AL vs NH residents) (lines 330-331). As suggested, we have added a brief comment about the potential role of less effective isolation of active cases and greater interactions among AL residents (lines 331-332); however, while more physically impaired, it would be inaccurate to state that most NH residents were bedbound. We also believe it would be misleading to assume that NH residents were exposed to fewer people than AL residents - given that similar visitor restrictions were implemented in both settings during the pandemic and the expected higher numbers of staff available in NH vs AL settings.

5. Did death and burden of comorbidities correlate? What about morbidity?

Response: We believe this comment relates to the likely associations between level of resident comorbidity and the outcome of hospitalization with death. For this outcome (as with all our outcomes), our models were adjusted for several covariates, including number of chronic conditions as noted in our Analysis section (see lines 181-183) and in the footnotes provided for Table 3. 

6. Did the admission and discharge criteria vary between waves, and between NH and AL?

Response: Thank you for this interesting question. As would be expected, there always have been differences in the admission and discharge criteria between licensed publicly subsidized AL and NH settings in Alberta as these criteria define initial and ongoing eligibility for setting-specific care as per Alberta Health Services (as per our reference #29). There is no evidence, however, that admission and discharge criteria changed in either of the two settings with the emergence of the COVID-19 pandemic (or by pandemic wave) (Auditor General of Alberta. COVID-19 in Continuing Care Facilities, 2023. https://www.oag.ab.ca/wp-content/uploads/2023/04/oag-covid19-cont-care-facilities-feb2023.pdf ). However, we can’t be sure whether there were temporary differences in how existing admission and/or discharge criteria across AL and/or NH sites were enacted by the homes during the pandemic waves under study. It is also possible that strategies were adopted during the pandemic (e.g., increased interactions between NH sites and ED physicians or paramedic services [Wyer et al., 2024]) that differentially affected hospitalization events during the four pandemic waves. We have added a brief comment regarding the above to our Discussion along with the references noted above (lines 343-348; new references #48 & #49).

Reviewer #2:

1. The article is well constructed and gives an interesting perspective of how different settings of assisted living and nursing homes respond during health crises.

Response: Thank you for these positive comments regarding the strengths of our manuscript.

2. Some data should be clarified and be less dispersive to ensure that readers undestand exactly the researches point of view.

Response: Thank you. We are unclear what specific data should be clarified and/or rearranged (which we think is what was meant by the suggestion to be less dispersive). We believe that our revisions made in response to comments provided by Reviewer #1 help to address this comment.

3. I did appreciate the strategies to be investigated in the conclusions and I think it worths finding out more about.

Response: Thank you for this positive comment.

---

## [Decision Letter · Decision Letter 1]

20 Jun 2024

Comparison of hospitalization events among residents of assisted living and nursing homes during COVID-19: do settings respond differently during public health crises?

PONE-D-24-03317R1

Dear Dr. Maxwell,

We’re pleased to inform you that your manuscript has been judged scientifically suitable for publication and will be formally accepted for publication once it meets all outstanding technical requirements.

Kind regards,

Peng Wu

Academic Editor

PLOS ONE

Additional Editor Comments (optional):

Reviewers' comments:

Reviewer's Responses to Questions

**Comments to the Author**

1. If the authors have adequately addressed your comments raised in a previous round of review and you feel that this manuscript is now acceptable for publication, you may indicate that here to bypass the “Comments to the Author” section, enter your conflict of interest statement in the “Confidential to Editor” section, and submit your "Accept" recommendation.

Reviewer #2: All comments have been addressed

Reviewer #3: All comments have been addressed

2. Is the manuscript technically sound, and do the data support the conclusions?

Reviewer #2: Yes

Reviewer #3: Yes

3. Has the statistical analysis been performed appropriately and rigorously? 

Reviewer #2: Yes

Reviewer #3: Yes

4. Have the authors made all data underlying the findings in their manuscript fully available?

Reviewer #2: Yes

Reviewer #3: Yes

5. Is the manuscript presented in an intelligible fashion and written in standard English?

Reviewer #2: Yes

Reviewer #3: Yes

6. Review Comments to the Author

Reviewer #2: (No Response)

Reviewer #3: (No Response)

7. PLOS authors have the option to publish the peer review history of their article (what does this mean?). If published, this will include your full peer review and any attached files.

Reviewer #2: No

Reviewer #3: No

---

## [Editor Report · Acceptance letter]

2 Jul 2024

PONE-D-24-03317R1 

PLOS ONE

Dear Dr. Maxwell, 

I'm pleased to inform you that your manuscript has been deemed suitable for publication in PLOS ONE. Congratulations! Your manuscript is now being handed over to our production team.

Kind regards, 

on behalf of

Dr. Peng Wu 

Academic Editor

PLOS ONE